# Addressing Challenges in Fabricating Reflection-Based Fiber Optic Interferometers

**DOI:** 10.3390/s19184030

**Published:** 2019-09-18

**Authors:** Markus Solberg Wahl, Øivind Wilhelmsen, Dag Roar Hjelme

**Affiliations:** 1Department of Electronic Systems, Norwegian University of Science and Technology (NTNU), 7491 Trondheim, Norway; dag.hjelme@ntnu.no; 2Department of Energy and Process Engineering, Norwegian University of Science and Technology (NTNU), 7491 Trondheim, Norway; oivind.wilhelmsen@ntnu.no; 3SINTEF Energy Research, 7034 Trondheim, Norway

**Keywords:** fiber optics, multimode interference, refractive index (RI) sensor

## Abstract

Fabrication of multimode fiber optic interferometers requires accurate control of certain parameters to obtain reproducible results. This paper evaluates the consequences of practical challenges in fabricating reflection-based, fiber optic interferometers by the use of theory and experiments. A guided-mode propagation approach is used to investigate the effect of the end-face cleave angle and the accuracy of the splice in core-mismatched fiber optic sensors. Cleave angles from high-end fiber cleavers give differences in optical path lengths approaching the wavelength close to the circumference of the fiber, and the core-mismatched splice decides the ensemble of cladding modes excited. This investigation shows that the cleave angle may significantly alter the spectrum, whereas the splice is more robust. It is found that the interferometric visibility can be decreased by up to 70% for cleave angles typically obtained. An offset splice may reduce the visibility, but for offsets experienced experimentally the effect is negligible. An angled splice is found not to affect the visibility but causes a lower overall intensity in the spectrum. The sensitivity to the interferometer length is estimated to 60 nm/mm, which means that a 17 µm difference in length will shift the spectrum 1 nm. Comparisons to experimental results indicate that the spliced region also plays a significant role in the resulting spectrum.

## 1. Introduction

Optical fiber interferometers based on cladding mode excitation have received much attention the last two decades due to their simple fabrication and robust, all-fiber design. The multimode interference created has been used to measure temperature [1], strain [2], refractive index (RI) [3,4] and as a fiber focusing lens [5]. The interferometric sensors can be divided into Mach–Zehnder type (transmission) geometries and Michelson type (reflection) geometries, or based on the method of cladding mode excitation (e.g., core diameter mismatch [1,6], core offset [4] or long-period gratings [7]). Sensitivity to the external RI is created by the evanescent field in configurations where the light is bound by the outer diameter of the fiber. With different types of stimuli-responsive coatings, this can also enable sensitivity to other parameters (e.g., relative humidity [7] and pH [8,9]).

Countless different sensor designs have been reported. The most common geometries have been reviewed by Zhu et al. [10], and Lee et al. [11]. The number of reports that addresses the deeper understanding of the multimode interference systems are limited (e.g., [5,12,13,14,15]), and the discussions in these papers are limited to circularly symmetric geometries. This is a reasonable assumption for many of the Mach–Zehnder geometries, while for the Michelson geometries this is often not the case due to practical limitations in the sensor preparations.

The objective of this paper is to study the impact of geometrical asymmetries on the performance of Michelson class of sensors. Full vector field model simulations, validated through experiments, are used to study the effects of fabrication induced asymmetries. A coreless fiber interferometer was chosen because it represents the simplest form of in-line interferometer, yet complex enough to illustrate the effects of imperfections in implementation. The purpose is to aid sensor development by providing better understanding of the impact of the non-ideal sensor geometries. This understanding combined with process control and repeatability are important factors to ensure the advancement and maturing of the technology.

The general advantages of fiber optic sensors are the inert material which enables sensing in chemically harsh environments; a small thermal mass which minimizes the influence on thermodynamic reactions; their immunity to electromagnetic interference; and their small size. Additionally, the inline interferometers based on cladding mode excitation are robust and easy to fabricate. The reflection-based geometry of the Michelson type interferometers also enables the fiber optic interferometers to be used as insertion probes. This may simplify the experimental implementation and reduce thermally induced strain and bending effects between the two anchoring points necessary for the transmission type geometry. Despite the advantages of a reflection-based geometry, most papers deal with in-line sensors in which interrogation is done in transmission [9,12,13,16,17,18,19,20,21]. The reason for this may be challenges associated with fabrication of the reflection-based sensors (e.g., the accuracy of the end-face cleave angle). A cleave angle that is not perpendicular to the fiber axis will affect the distribution of cladding modes as it eliminates the radial symmetry of the systems and removes the translational invariance in the propagation direction, often assumed in the treatment of such systems [5,12]. The experience from our studies is that interferometers fabricated with the same intended dimensions and conditions, will exhibit different interference spectra.

High-end fiber cleavers typically claim that the cleave angles are within 0.3 degrees [22]. Whereas this is insignificant for the core-bound mode of a single mode fiber (SMF), cladding modes with field distributions closer to the circumference of the fiber can experience optical path length differences approaching 1 µm from one side to the other.

This paper examines, by experiments and theory, the effects of the spliced region and the end-face cleave angle, through a guided-mode propagation analysis (MPA). The study is limited to geometries with the original cladding diameter (i.e., it will not cover reduced diameter sensors based on etching with hydrofluoric acid). Linear and lossless media are assumed.

Previous papers based on MPA have treated low index contrast cases and weakly guiding conditions, where a scalar (beam optics) treatment is sufficient [5,12,13,15,23,24]. In this work, a full (electromagnetic) vector model will be used. The more general vectorized model is chosen for several reasons: Only the fundamental mode will experience a low index contrast in a single-mode fiber, the cladding modes are subject to the medium outside the fiber, which lifts the degeneracy of the LP approximation. The model also accounts for the mixed polarization of the hybrid modes and ensures that the contribution from all is utilized, also when the system lacks radial symmetry. This study will treat coreless (CL) fiber interferometers, but the model is applicable to all core-mismatch geometries. The next section will introduce the model on which the simulations are based. The rationale for the paper will then be explained with experimental results, before these are analyzed based on results from the simulation.

## 2. Methods

The sensor is fabricated by splicing a section of coreless (CL) fiber FG125LA (Thorlabs) to a single-mode fiber (SMF-28) pigtail, see Figure 1. The difference in numerical aperture between the two fibers enables the excitation of multiple modes in the CL fiber, which create the interference spectrum. A silver mirror is chemically deposited at the end-face through the mirror reaction. The reactant solution was prepared according to [25] and the tip of the fiber was placed in contact with the liquid. The fiber was withdrawn carefully without breaking the meniscus created by surface tension. This yielded a silver layer of an approximate thickness of 30 µm, and with minimal deposition on the side of the fiber. With a 2 × 1 coupler (Huber Suhner), a broadband source (Fyla, SCT500), and a grating-based spectrometer with 0.3 nm resolution (Ibsen I-MON 512), the multi-mode interferometer is interrogated. The length of the SMF between the source, detector and sensor, ensures that only the fundamental mode of the fiber can be assumed to be transmitted. Therefore, only the region from the splice to the end-face of the fiber is considered in the model.

### Guided-Mode Propagation Model

The components of the electromagnetic field can be written in terms of the radial $\left(r\right)$, azimuthal $\left(\varphi \right),$ and longitudinal $\left(z\right)$ variables:(1)$$\Psi_{lm}\left(r,\varphi ,z\right)=\psi_{lm}\left(r\right)e^{-jl\varphi }e^{-j\beta_{lm}z}$$

$\;\psi_{lm}$ is the electric E− or magnetic H-field profiles, which are always real for non-absorbing fibers [27]. The field $\psi_{lm}\left(r\right)$ has radial indices m and azimuthal indices l and propagate with propagation constants $\beta_{lm}$ along the z-axis. $\psi_{lm}\left(r\right)$ is predominantly transversal with negligible longitudinal components.

The input (SMF) field $\psi_{lm}$ is calculated through a finite-element method with the parameters given in Table 1. The convergence criterium was $\Delta n_{eff}<10^{-9}$. The CL mode fields are found in the same way and classified based on their indices l and m. The modes with m≤15 and l≤4 were included in the simulations. This is not the full set of hybrid modes that the fiber could support, but includes the modes that may be significantly excited by the core diameter mismatch. The excitation coefficients of the hybrid modes are calculated individually, however, only the sum of the square magnitude for each value of l and m is presented. The set of modes are mutually orthogonal [28] and were normalized to create an orthonormal set [27].

The abrupt change in core diameter at the splice enables the excitation of cladding modes in the CL fiber. The reflection at the SMF-CL interface is small and is ignored. This was confirmed experimentally in [10]. Continuity at the interface then requires that the input transverse field from the SMF ($E_{t,SMF}$) is equal to the sum of excited modes and the radiative fields in the CL section [27] (p. 420):(2)$$E_{t,SMF}\left(r,\varphi \right)=\underset{l,m}{\sum }a_{lm}e_{t,lm}\left(r,\varphi \right)+E_{t,rad}\left(r,\varphi \right)$$

Here, $e_{t,lm}$ and $a_{lm}$ are the transverse portion of the mode fields and modal amplitude with indices l and m. The radiation field will in most cases not reach the detector and is therefore omitted in the calculations. Neglecting the wavelength dependency, the modal amplitudes $a_{lm}$ in the excited field are found by calculating the overlap integrals
(3)$$a_{lm}=\frac{1}{2N_{lm}}\int_{A}E_{SMF}\times h_{lm}^{*}\cdot \hat{z}dA$$
(4)$$\;N_{lm}=\frac{1}{2}\left|\int_{A}e_{lm}\times h_{lm}^{*}\cdot \hat{z}dA\right|$$

Due to the core mode of the SMF lacks azimuthal components, $a_{lm}$ is zero for all modes where l≥1 (i.e., only radial $HE_{1m}$ modes will be excited), when the splice is radially symmetric. However, if the fiber is cleaved at an angle, the symmetry is broken and azimuthal modes may be excited [27]. The effect of the angled end-face is to produce a linear phase shift across the mode and a spatial displacement between the incident and reflected mode. For small cleave angles the effect of the displacement is negligible. Thus, the effect of the angled end-face is modelled by propagating the initially excited set of modes to the cleaved interface, where the field becomes:(5)$$e_{cleave}=\sum a_{lm}e_{t,lm}e^{-j\beta_{lm}z\left(r\right)}$$
where $z\left(r\right)=z_{0}+2\theta r\cos\varphi$ for small cleave angles θ, over the fiber radius r and azimuthal angle φ, as defined in Ref. [27] and Figure 1. $z_{0}$ is the original length of the fiber. The excited modes in the reflected field can similarly be found with the overlap integral between $e_{cleave}$ and the mode fields:(6)$$b_{lm}=\int e_{cleave}\times h_{lm}^{*}\cdot \hat{z}dA$$
where l≥1 modes are included. This corresponds to having an interface of perfect reflectivity, independent of the field polarization. The latter is valid provided that the cleave angle is small. This is conceptually equivalent to the transmission from one fiber to another that is tilted, as described in [27].

After propagating the reflected field back to the SMF, the total magnetic field at the splice is then expressed as
(7)$$h_{refl}=\sum_{lm}b_{lm}h_{t,lm}e^{-j\beta_{lm}z}$$

The power transferred back to the SMF then becomes:(8)$$P_{out}={\left|\frac{1}{2}\int_{A}E_{SMF}\times h_{refl}^{*}\cdot \hat{z}dA\right|}^{2}$$

The output spectrum is hence a function of only the effective RI and the relative contribution of each mode. The splice between the SMF and the CL fiber, and a potential angled end-face cleave, may therefore affect the spectrum shape. The inaccuracies of the splice are introduced by altering the input field $E_{SMF}$ in (3).

To calculate the effective indices over the wavelength interval considered, the modes are calculated at two wavelengths to account for material [26] and waveguide dispersion. A linear approximation is employed to find the effective index for each mode as a function of wavelength. This approximation is considered valid as the derivative of the effective index with regards to wavelength is approximately constant [14].

Since the evanescent field is present around the fiber, a change in the external RI will affect the effective RI of the modes. To incorporate this RI sensitivity in the model, the confinement coefficients are calculated from the mode fields, as a function of wavelength. The effective RIs of the modes, as a function of external RI, are then estimated from a weighted average based on the confinement coefficients. The model does not, however, consider the decreased confinement of the modes caused by the lower index contrast at higher external RI. According to the study [14], this seems to be a reasonable approximation. The mode fields used to calculate the excitation coefficients are the same for all wavelengths. Thus, the effect of mode field changes with wavelength, resulting in a small and uniform change in the excitation coefficients, are considered negligible.

## 3. Experimental Results

The treatment of the various challenges that may arise when fabricating fiber-optic interferometers starts with Figure 2, which shows the spectra from six sensors fabricated through the exact same process. The sensors consist of 14.2 mm of CL fiber and are denoted by the measured end-face cleave angle. The length was chosen because it puts a characteristic feature within the desired wavelength interval that is easily recognizable in the otherwise chaotic spectrum. The characteristic feature is the zero intensity range from around 1570 nm and upwards. Another option was the re-imaging length of 29.8 mm [5], but limitations of the cleave-length measurement prevented this.

The fusion splicer measures the angle with an accuracy that is not disclosed by the manufacturer. Subsequent measurements of the same cleaved fiber have been found to give angles differing by ±0.1 degrees. The angles in the figure are therefore included for guidance only. Moreover, the angle could not be controlled, only measured after fabricating the fibers. Many sensors were therefore fabricated to find a set of sensors with the desired approximate cleave angles.

To optimize the accuracy of the interferometer length, a DC servo motor was used to pull back the spliced region, the intended length from the cleaver blade. The spectra still had to be shifted up to 0.8 THz (i.e., 6 nm at 1500 nm), to align well, as shown in Figure 2. After correcting for the length inaccuracies, the spectra can be compared. For small cleave angles, the different realizations of the same geometry give the same general trend, but with two main differences: The spectra may be shifted in wavelength and the relative amplitude of the resonances may differ.

The measured RI sensitivity in one of the sensors was measured to 63.3 nm/RIU at $n_{ext}$ = 1.33, and an average of 146 nm/RIU on the interval 1.33–1.39.

Only the cleave angle is measured, but other variabilities may contribute. This motivates an analysis through simulations where the different sources of variabilities can be isolated. To further understand these results, simulations were performed with the guided-mode propagation model presented in Section 2. The next section will present the general findings from the simulations, with separate investigations of each source of error. This will form the basis for discussions of the complete system.

It was decided to show linear y-axes to highlight the constructive interference peaks which are more robust. The low intensity changes of dips ($10^{-3}-10^{-4}$) are difficult to detect experimentally and very sensitive to the numerically derived results.

## 4. Simulation Results

The simulation results for a CL fiber interferometer will be presented next to analyze the effect of the broken radial symmetry of the fiber. The spectra and excitation coefficients for modes with azimuthal indices l≤1 will be shown. Modes with l≥2 were also excited, but with coefficients an order of magnitude smaller. These are not shown due to their small contribution.

Constructive interference occurs when the difference in the reflected optical pathlength between the two modes is equal to an even number of π, which gives [29]:(9)$$\lambda_{peak,i}=\frac{4L\left[n_{lm}^{eff}\left(\lambda ,n_{ext}\right)-n_{l{}^{\prime }m{}^{\prime }}^{eff}\left(\lambda ,n_{ext}\right)\right]}{2i}$$
where $n_{lm}^{eff}$ and $n_{l{}^{\prime }m{}^{\prime }}^{eff}$ are the effective RI of the two modes with indices l and m, at the wavelength λ and external RI ($n_{ext}$). L is the interferometer length and i is an integer. The change in resonance peak wavelength due to changes in external RI and/or length can be derived from the total differential of Equation (9):(10)$$d\lambda_{peak}=\frac{4L}{2i}\left(\partial_{\lambda }n_{lm}-\partial_{\lambda }n_{l{}^{\prime }m{}^{\prime }}\right)\;d\lambda_{peak}+\;\frac{4L}{2i}\left(\partial_{n_{ext}}n_{lm}-\partial_{n_{ext}}n_{l{}^{\prime }m{}^{\prime }}\right)\;dn_{ext}+\frac{4}{2i}\left(n_{i}-n_{j}\right)\;dL$$
where $\partial_{x}$ refers to the partial derivative with regards to x. By rearranging and substituting in the group index $\left(n_{g}=n-\lambda \frac{\partial n}{\partial \lambda }\right)$, and keeping the other variables constant, the sensitivity to length becomes:(11)$$\frac{d\lambda_{peak}}{dL}=\frac{\lambda_{peak}}{L}\frac{n_{lm}^{eff}-n_{l{}^{\prime }m{}^{\prime }}^{eff}}{n_{lm}^{g}-n_{l{}^{\prime }m{}^{\prime }}^{g}}$$
for the same modes. The linear chromatic dispersion assumed in the model causes constant group indices with regards to wavelength, where higher order modes have higher group indices. This causes the nominator and denominator in the last fraction to have opposite signs, which gives a negative sensitivity to length variations.

The RI sensitivity from interference between the two modes can in the same way be expressed as:(12)$$\frac{d\lambda_{peak}}{dn_{ext}}=\frac{\lambda_{peak}}{\left(n_{lm}^{g}-n_{l^{\prime }m^{\prime }}^{g}\right)}\left(\frac{\partial n_{lm}}{\partial n_{ext}}-\frac{\partial n_{l^{\prime }m^{\prime }}}{\partial n_{ext}}\right)$$

In a CL fiber, the sensitivity will be positive for all cladding modes [14], but may become negative in systems where a core bound mode is more strongly excited [4]. References [15,23,24] all show a positive sensitivity—for both CL and thin-core fibers. It should also be mentioned that the RI sensitivity has been found to vary little with the interferometer length [19,24], which has also been observed throughout this work. This is expected from (12).

### 4.1. General Results

The simulated spectra for different lengths are shown in Figure 3a. The peaks blueshift 60 nm/mm at 1500 nm (i.e., a 17 µm longer interferometer shifts the dip one nanometer). This means that to ensure a reproducibility of the dip wavelength of around 0.1 nm, the length needs to be controlled in the µm-range. The same sensitivity is found by inserting the effective indices into (11). This means that the length accuracy in Figure 2 is ±100 µm, which was also found in [8].

Larger variations in the length will change the typical period of the oscillations in the spectrum, as shown in [13]. With a longer propagation distance, the difference in phase delay from the small variations in effective index has time to differentiate (i.e., a longer interferometer exhibits higher frequency oscillations).

Figure 3b shows the excitation coefficients that together with the effective indices of the modes create the spectra shown in Figure 3a and Figure 4, calculated with (3). The coefficients for a straight cleave are consistent with what was found in [24].

The simulated RI sensitivity of the 14.2 mm CL interferometer in this work was found to be 65.5 nm/RIU, as shown in Figure 4, which is lower than what is reported elsewhere [15,23,24]. Compared to our experimental results, the simulation is valid only for RIs close to $n_{ext}$ = 1.33, since the model does not consider the increased evanescent field at higher RIs. The model considered in this work is expected to underestimate the RI sensitivity as it assumes that the mode fields are independent of the external RI. The mode fields are less confined at longer wavelengths, which increases the effect of the external RI on $n_{lm}^{eff}\;\left(\lambda ,n_{ext}\right)$, although the effect is small in the narrow wavelength range considered here.

### 4.2. Cleave Angle

The total field profile changes with the propagation distance, as shown in [12]. With a straight cleave, the fiber profile is identical both before and after the reflection, and the reflected field will remain unchanged. However, when the end-face cleave is not perpendicular to the fiber axis the symmetry is broken and non-radial modes can be excited. The excitation coefficients now depend on the optical path length from the splice to the cleaved end-face (i.e., the accumulated phase). The coefficients for a CL interferometer cleaved at moderate angles are shown in Figure 5, where the coefficients are calculated according to (6) at the fiber end-face. The lengths and wavelengths chosen were the re-imaging length (29.8 mm) at 1550 nm, 14.2 mm at 1550 nm, and 14.2 mm at 1216 nm. The re-imaging length was identified from the simulations by finding the length and wavelength where the input field reproduced itself. This length was chosen to clearly illustrate the effect of an angled cleave, when all modes are in phase. The coefficients for a 14.2 mm interferometer (Figure 5c–f) show how the ensemble of modes excited at the fiber end-face change as a function of the cleave angle, according to (5) and (6). The excitation coefficients for the two wavelengths behave differently because the phase of each mode varies independently with its propagation constant. In all cases, the coefficients for a straight cut are identical to the initial set of modes excited at the SMF-CL interface, as shown in Figure 3b. However, as the end-face is tilted slightly, the energies of the radial modes decrease. Concurrently, the tilted phase front of the incoming field enables modes with azimuthal components to increase their contribution, as shown in Figure 5b,d,f. The higher order azimuthal modes (l>1) also increase, but with a contribution an order of magnitude lower.

Figure 6 shows interference spectra for a 14.2 mm CL interferometer with different cleave angles. The interference amplitude can be seen to decrease as the angle increases—in the range 1400–1500 nm, the interferometric visibility is reduced by 20–70% when the angle is increased from zero to 0.3 degrees. The l = 1 modes are orthogonal to the single mode of the input fiber and will not contribute to the observed interference spectrum. Thus, the tilt introduces a loss mechanism in the sensor, reducing the intensity of the interference signal. The effect of the cleave angle on the RI sensitivity is shown in Table 2.

### 4.3. Splice Offset

The inaccuracy of the splice affects the coupling of modes both into and out of the interferometer. The effect of an offset splice was examined by shifting the incoming SMF field along one axis. Intentional splice offsets to excite cladding modes have been studied experimentally in [4]. The effect of inaccuracy in the splice is illustrated in Figure 7. The overall shape is less affected, which follows from the smaller change in excitation coefficients. Modes with l>1 are also excited in this case, but with excitation coefficients an order of magnitude lower, even at large offsets. This is expected, as the field of the lower order modes in the CL fiber are broad (i.e., extends the full fiber cross-section as indicated in Figure 1). This will reduce the effect of a small offset of the fundamental SMF mode, which is confined to the center of the fiber. Contrary to the angled cleave, an offset seems to lower the intensity of the broad peak in the 1500–1575 nm spectral region.

As the result for an angled cleave, the effect of a small offset also reduces the visibility of the spectrum resonances, but to a lesser degree (i.e., 3–25% in the range 1400 to 1500 nm), for the largest offset simulated. Experimentally, the offsets have in general been measured close to zero, and not larger than 0.5 µm. This means that the isolated effect should be negligible.

As each mode has a different RI sensitivity, the total sensitivity of the interference could be altered by an angled cleave or offset splice. However, according to the simulation results, the effect is almost negligible with only a slight increase for an angled cleave, see Table 2 below. As the dip visibility is reduced, the determination of the wavelength becomes less accurate. The depth of the dips decreased by almost two orders of magnitude, but how this effect appears experimentally may be partly washed out by the noise floor of the detector, the linewidth of the light source or the wavelength resolution of the detector.

### 4.4. Splice Angle

If the two fibers are spliced at an angle, the incoming phase front is tilted. As explained earlier, this enables azimuthal modes to be excited. The splice angles have been exaggerated in Figure 8 to highlight the effect. The spectra show that the overall intensity is reduced if the fibers are spliced at an angle. However, the interferometric visibility is maintained in the interval 1400–1500 nm. That the effects are less prominent than for an angled cleave is expected, as the input field is confined to the center of the fiber where the tilt-induced difference in optical pathlength is less than closer to the circumference of the fiber. Splice angles measured experimentally have not exceeded 0.2 degrees, which according to the model would not cause a significant effect.

## 5. Discussion

### 5.1. Comparison of Experiment and Simulations

The simulated spectra of the 14.2 mm interferometer are shown in Section 4. Compared to Figure 2 it seems like the long wavelength cut-off/intensity drop is 40 nm red-shifted in the experimental results. This corresponds to a 0.67 mm shorter propagation length or a lower RI than what is used in the simulations. An underestimated interferometer length may be a consequence of the real-life beam expansion occurring at the hetero-interface.

For small cleave angles (≤0.2 degrees) there appears to be no correlation between the magnitude of the cleave angle and the observed spectral change. This is surprising as a nonzero cleave angle will increase the insertion loss (integrated across a wide wavelength range compared to the free spectral range of pairwise mode interference) of the device, resulting in a monotone decrease in intensities of the interference peaks as seen in the simulations. The insertion loss is due to the mode coupling form radial modes to azimuthal modes. The insertion loss estimated from the spectra in Figure 2 shows a decrease with the cleave angel varying from zero to 0.2 degrees. This might be explained by the inaccuracy of the cleave angle measurement and by the nonideal SMF-CLF splice (offset and/or angle) that will introduce mode coupling between the azimuthal modes and the SMF mode.

The periodicity, or the variations in periodicity, are comparable in the range 1350–1600 nm, although the amplitude of the different peaks varies. In the range 1450–1550 nm, the experimental results for low cleave angles show decreasing oscillations for lower wavelengths. Comparing with the simulations, this is a feature that is more prominent in the spectra for higher cleave angles, whereas the inaccuracies of the splice show the opposite effect. When the experimental spectra exhibit this behavior independent of the measured angle, a reasonable assumption would be that the effect is caused by the same mechanisms (i.e., the effect shown in Figure 5). A different distribution of power into the various modes changes the relative amplitudes of the peaks. It should also be noted that the excitation coefficients may show some variation with the wavelength, which is not considered in the model.

If the measured angle is approximately correct, the effect must stem from the spliced region. The simulations show that the end-face cleave angle causes the greatest effect on the spectrum, but this part of the sensor is also the part best described by the model. The splice is shown to have less effect in the simulations, but this part is not as well represented in the model. In a fusion splicer, the two fibers are pushed together with a small overlap, which creates a thicker region where the core geometry is less defined. It may also not be radially symmetric, as assumed in the model. This may excite more non-radial modes than what the model indicates and enable the azimuthal modes to contribute in the measured spectra. This would create spectra more resembling the experimental results. The abrupt coreless interface may therefore be a less accurate description. It seems like the non-abrupt splice may be less efficient in exciting the modes than the simulation shows.

### 5.2. The Electromagnetic Vector Model

The model used in this work is based on vectorized mode fields and their corresponding propagation constants. The rationale for the use of this model was the lifted degeneracy of the modes in systems with higher index contrast and the lack of radial symmetry in inaccurately fabricated sensor geometries. The four modes for each value of l>0 are of interest, as they have individual excitation coefficients. The longitudinal components of the mode fields also increase at higher index contrast, but these disappear in the multiplication of the cross-product with the z-vector in (3).

As many of the modes are not linearly polarized (i.e., the polarization is a function of the position in the fiber), the effect of tilting the end-face is difficult to analyze with a scalar model. The individual contribution for LP modes is not well defined. The general model developed here is also suited for expanding the investigation into new systems with different geometries.

The RI sensitivity was found to vary little with the fabrication inaccuracies that were investigated with the MPA model. This is expected from the excitation coefficients, which show that the same modes contribute to the spectra. The results also confirm the experimental sensitivity around an RI of 1.33. However, because the change in the evanescent field with refractive index change is not described in the modes, this causes the underestimation found in the simulated RI sensitivity at higher RIs.

The vector model confirms that only the radially symmetric $HE_{1m}$ modes need to be considered in radially symmetric systems. It may, however, be difficult to guarantee perfect symmetry in an experimental setting. It seems like the excitation of l≥1 modes can reduce the visibility of strong destructive interference dips originally present. According to the simulations, it is the cleave angle that has the largest impact on the spectrum. This may be understood from the large amount of modes present at the cleaved end-face, of which many have field distributions that span far out from the fiber center. This will increase the coupling into higher order azimuthal modes in the reflected field, as shown in Figure 5. That the effect is an interference phenomenon can be seen from each of the modes being affected differently. At each wavelength, the excitation coefficients exhibit different behavior, according to the phase relation of the incoming modes. Experimentally, however, the results indicate that the splice may affect the result to a larger degree.

## 6. Conclusions

The challenges of fabricating reflection-based, core-mismatched fiberoptic sensors based on multimode interference have been investigated by simulations and experiments. The mode-field propagation approach was used to explore the effect of an end-face cleave angle, an offset fusion splice, and an angled fusion splice. Whereas the fiber geometry defines the available modes and their propagation constants, the interfaces in the system decide the power distribution among the modes.

The experimental results showed the variability in the spectrums produced from a set of sensors with different cleave angles. Some peaks showed a consistent increase in intensity for larger cleave angles, whereas other showed a decrease. The length accuracy of the experimental setup was found to be ±100 µm based on the results from the simulations, which gave an estimated sensitivity to the interferometer length of 60 nm/mm. The refractive index sensitivity was estimated to be 65.5 nm/RIU.

According to the simulations, the end-face cleave angle has the largest impact on the spectrum, attributed to the wide distribution of optical power in the cross-section. The interferometric visibility was seen to be reduced by up to 70%. The inaccuracies in the spliced region were less prominent in the simulations, which was explained by the confined SMF mode field being less affected by the small alterations in the CL field. The reduction in visibility from an offset splice was less than 25% for the large offsets simulated, but for offsets experienced experimentally the effect becomes negligible. The angled splice did not affect the visibility but caused an overall lower intensity in the spectrum. However, the experimental results resembled the angled cleave simulations, but the effects were not always consistent. This indicates that the actual splice creates mode excitation similar to the effect of an angled cleave, with a larger portion of azimuthal modes excited.

This paper was written in the prospect that awareness of the challenges and mechanisms involved will enable more reliable and reproduceable sensors in the future.

## Figures and Tables

**Figure 1 sensors-19-04030-f001:**
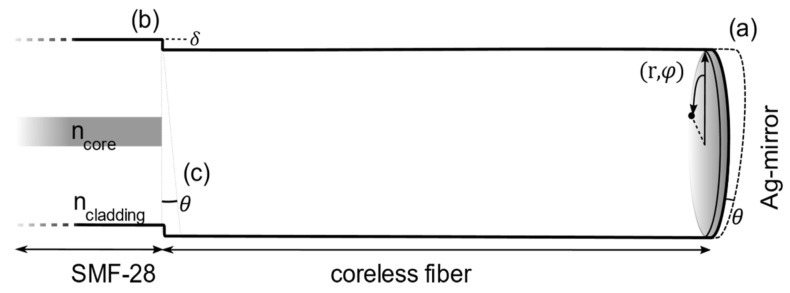
Schematic configuration of the challenges associated with the fabrication of fiberoptic interferometers based on core mismatch. The three parameters investigated are an end-face cleave angle (**a**), a lateral offset in the splice (**b**) and a tilted splice (**c**), as indicated in the figure.

**Figure 2 sensors-19-04030-f002:**
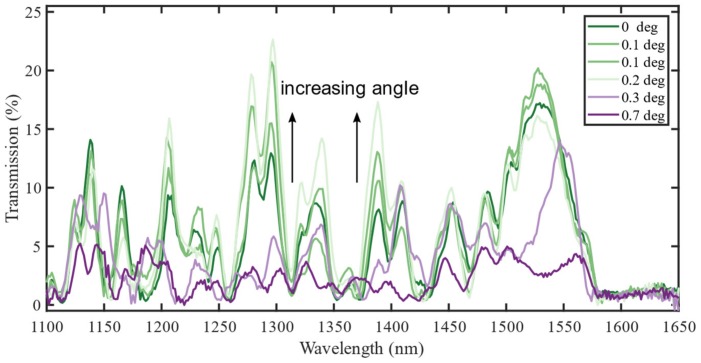
Experimentally measured spectra of a 14.2 mm coreless (CL) interferometer. The spectra (0.2 and 0.3 degrees) are shifted 6 nm to align better for comparison. This corresponds to a cleave length accuracy of ±70 μm according to the results in this paper.

**Figure 3 sensors-19-04030-f003:**
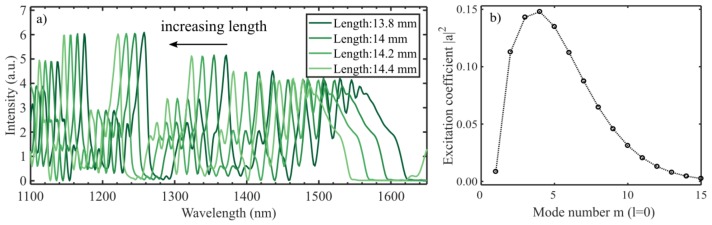
(**a**) Spectra of coreless interferometers with different lengths that have a perfect cleave. The spectrum sensitivity to the length of the interferometer is 60 nm/mm at around 1500 nm. (**b**) Excitation coefficients for the radial modes $\left(l=0\right)$ at the single mode fiber (SMF)-CL fiber interface. The excitation of the hybrid modes is calculated individually, however, only the sum of the square magnitude is plotted for each value of l.

**Figure 4 sensors-19-04030-f004:**
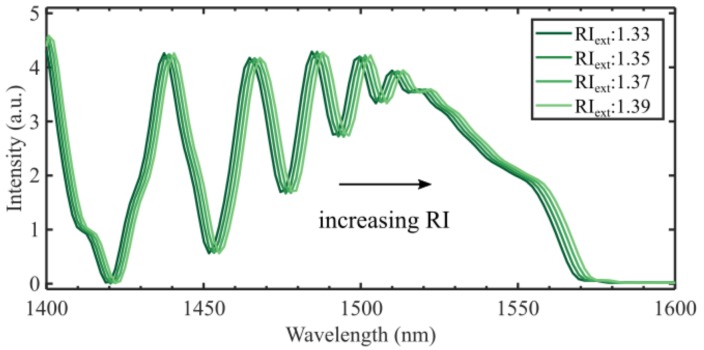
Refractive index sensitivity for a 14.2 mm CL fiber with a perfect cleave (zero degrees). The resulting sensitivity is 65.5 nm/RIU at 1550 nm.

**Figure 5 sensors-19-04030-f005:**
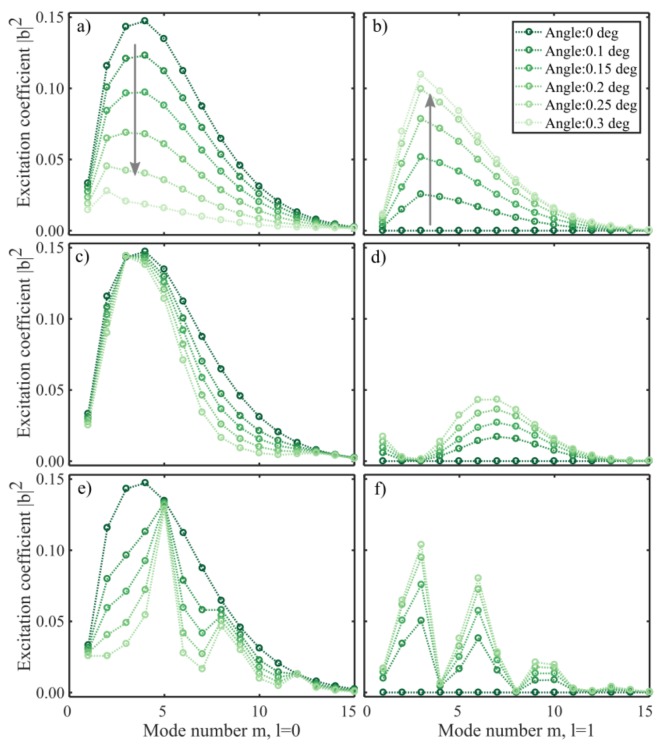
Excitation coefficients (Equation (6)) at the 1550 nm re-imaging length (29.8 mm) (**a**,**b**), at 1550 nm (14.2 mm), and at 1216 nm (14.2 mm), as a function of the cleave angle. The excitation of radial modes (l=0) decrease (**a**) as the power of the azimuthal modes $\left(l=1\right)$ increase (**b**,**d**,**f**)).

**Figure 6 sensors-19-04030-f006:**
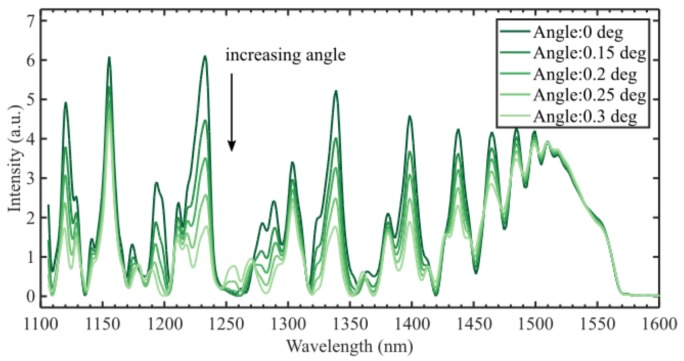
Simulation of the CL interferometer with different cleave angles shows reduced visibility of the interference. In the range 1400–1500 nm, the decrease is 20–70% for 0.3 degrees cleave angle.

**Figure 7 sensors-19-04030-f007:**
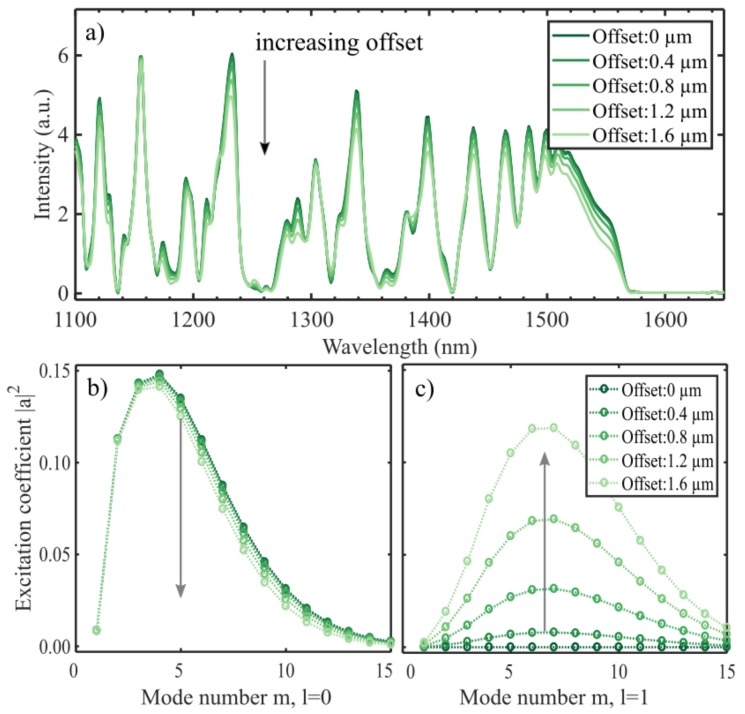
The output spectra for a 14.2 mm CL interferometer for moderate splice offsets (**a**). The peak intensities are in general reduced, whereas dips remain mostly unchanged. The excitation coefficients (**b**,**c**) are calculated at the splice between the SMF and the CL fiber (Equation (3)) for l=0 and l=1. With a larger offset more power is going into the azimuthal modes (**c**).

**Figure 8 sensors-19-04030-f008:**
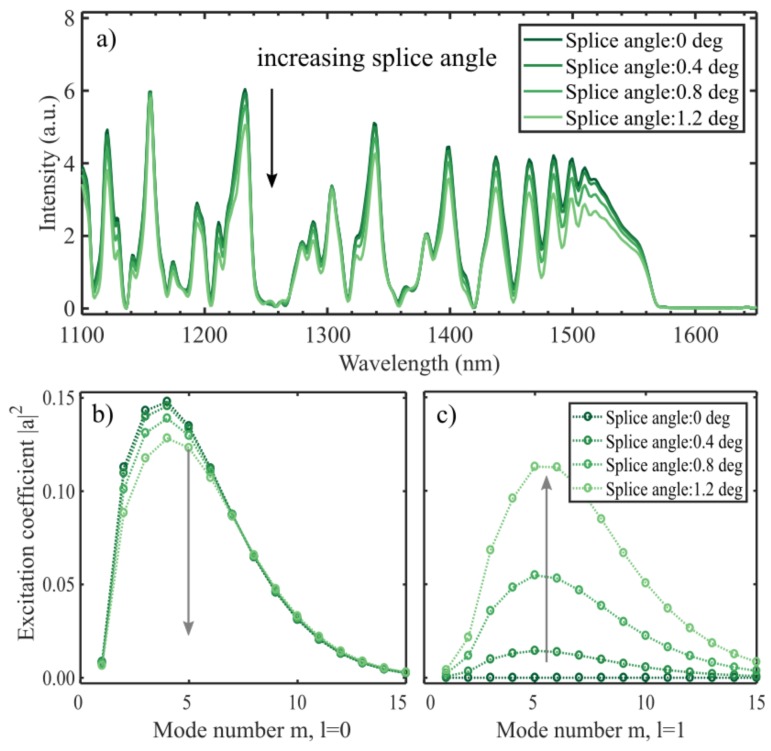
The spectra from a 14.2 mm CL interferometer which is spliced to the input fiber at an angle (**a**). To show the effect, the angles used are larger than what can be expected experimentally. An angled splice causes a lower overall intensity, as well as a small reduction in visibility of the interference. The excitation coefficients (**b**,**c**) are calculated at the splice between the SMF and the CL fiber (Equation (3)) for l=0 and l=1. With a larger offset more power is going into the azimuthal modes (**c**).

**Table 1 sensors-19-04030-t001:** Simulation parameters at 1550 nm. Material dispersion calculated from [26].

	SMF	Coreless
Core diameter (µm)	8.2	
Cladding diameter (µm)	125	125
$n_{core}$	1.4504	
$n_{cladding}$	1.4447	1.4447
$n_{ext}$	1.33	1.33

**Table 2 sensors-19-04030-t002:** Refractive index sensitivity for different inaccuracies, calculated at 1550 nm.

0 Deg. Angle	0.2 Deg. Angle	0.3 Deg. Angle	1.6 µm Offset
65.5 nm/RIU	65.6 nm/RIU	65.7 nm/RIU	65.6 nm/RIU

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
