# Peer review of "Addressing Challenges in Fabricating Reflection-Based Fiber Optic Interferometers"

_sensors, 2019, doi:10.3390/s19184030_

Round 1

Reviewer 1 Report

The paper “Addressing Challenges in Fabricating Reflection-Based Fiber Optic Interferometers” simulated the Michelson fiber-optical interferometer spectra affected by the sliced angle at the end of the fiber tip, and the lateral offset in the spice and titled splice. The research results are useful for researchers working on interferometer based fiber optical sensors, and can be published with the major revisions as below:

The authors should indicate the significance and advantages of the Michelson interferometer selected here, and why a coreless fiber is spliced at the end of the fiber tip. In section 2, a lot of experimental details are missing. For example, what is the fiber brand, what is the coupler purchased from, and what is the thickness of the silver mirror, how is the silver deposited? What is the broadband source brand and what is the spectrometer brand for detection? What is the resolution of the spectrometer? The 2X1 coupler defects will affect the experimental results too. In Figure 2, why the experimental results for lighting transmission versus the end-face cleave angle cannot match the simulations in Figure 6, and the variation trend is even contradictory? Please explain. Please add a reference for equation (9). Please explain how equation (9) is derived to be equation (10), or add a reference for it. Please add details on how equation (11) is derived from equation (9). Is the cleave angle in Figure 4 used for simulation 0 degree? If so, please indicate in Figure 4 caption. In Figure 5, what is the re-imaging length? How is the re-imaging length identified? The sensitivity of the interferometer at around 65.5nm/RIU should be experimentally tested, instead of just doing the simulations. The modes used in the simulation figures shall be listed.

Reviewer 2 Report

A guided mode propagation analysis (MPA) using a full electromagnetic vector model was exploited to investigate the transmission spectrum of the coreless fiber interferometer with a reflective structure. The effect of the spliced region or the end face cleave angle on the performance of the proposed interferometer was discussed. Since the numerical apertures of modes are predominantly determined by their effective indices, mode excitation depending on the cleaving angle of coreless fiber must be changed. It should be effective if the interference regarding the numerical aperture of modes with variations in the cleaving angle of coreless fiber is analyzed additionally. It is not easy to understand results in the article because all parameters are mutually interacted to induce the interference pattern of the proposed interferometer. In Fig. 3, what is the cleaving angle of coreless fiber? The length of coreless fiber is not provided in Fig. 3(b). What is the dependence of mode excitation on the length of the coreless fiber? The mode excitation at the interface between a single-mode fiber (SMF) and a coreless fiber can be confirmed if the optical spectrum of the proposed interferometer is converted by the fast Fourier transform. In Fig. 5, mode excitation is changed by the cleaving angle. The length of coreless fiber, however, can simultaneously affect mode excitation and the number of modes. The manuscript should be revised by an English native speaker. I do not recommend that such facts warrant publication in a journal of the quality and the standing of Sensors.

Round 2

Reviewer 1 Report

The manuscript is much improved at current version, so I recommend its publication in present format. Just one small typo I have noticed in the abstract, “Comparisons to experimental results indicates that” should be “Comparisons to experimental results indicate that”.

Reviewer 2 Report

It is not evident if the modal dispersion is considered to achieve the effective indices of optical modes. All discrepancy may be caused by the modal dispersion. The cleaving angle vs. numerical aperture of modes regarding the mode order must be included in the manuscript in order to discuss the mode excitation at the angled end face.

Round 3

Reviewer 2 Report

I recommend the revised manuscript be accepted for publication in Sensors.